# Large-Scale Distributed Learning via Private On-Device Locality-Sensitive Hashing

**Tahseen Rabbani**[*]
Department of Computer Science
University of Maryland
trabbani@umd.edu

**Marco Bornstein**[*]
Department of Computer Science
University of Maryland
marcob@umd.edu

**Furong Huang**
Department of Computer Science
University of Maryland
furongh@umd.edu

## Abstract

Locality-sensitive hashing (LSH) based frameworks have been used efficiently to select weight vectors in a dense hidden layer with high cosine similarity to an input, enabling dynamic pruning. While this type of scheme has been shown to improve computational training efficiency, existing algorithms require repeated randomized projection of the full layer weight, which is impractical for computational- and memory-constrained devices. In a distributed setting, deferring LSH analysis to a centralized host is (i) slow if the device cluster is large and (ii) requires access to input data which is forbidden in a federated context. Using a new family of hash functions, we develop one of the first private, personalized, and memory-efficient on-device LSH frameworks. Our framework enables privacy and personalization by allowing each device to generate hash tables, without the help of a central host, using device-specific hashing hyper-parameters (*e.g.* number of hash tables or hash length). Hash tables are generated with a compressed set of the full weights, and can be serially generated and discarded if the process is memory-intensive. This allows devices to avoid maintaining (i) the fully-sized model and (ii) large amounts of hash tables in local memory for LSH analysis. We prove several statistical and sensitivity properties of our hash functions and experimentally demonstrate that our framework is competitive in training large-scale recommender networks compared to other LSH frameworks which assume unrestricted on-device capacity.

## 1 Introduction

Locality-sensitive hashing (LSH) has proven to be a remarkably effective tool for memory- and computationally-efficient data clustering and nearest neighbor search [6, 15, 2]. LSH algorithms such as SimHash [6] can be used to search for vectors in collection $W \subset \mathbb{R}^d$ of massive cardinality which will form a large inner product with a reference vector $x \in \mathbb{R}^d$. This procedure, known as maximum inner product search (MIPS) [27], has been applied to neural network (NN) training. In NN training, the weights of a dense layer that are estimated to produce a large inner product with the input (thereby, a large softmax, for example) are activated while the remainder is dropped out.

While LSH-based pruning greatly reduces training costs associated with large-scale models, popular frameworks such as SLIDE [8] and Mongoose [7] cannot be deployed in distributed settings over

---

[*]These authors contributed equally to this work.

37th Conference on Neural Information Processing Systems (NeurIPS 2023).

memory-constrained devices such as GPUs or mobile phones for the following reasons: **(a)** required maintenance of a large target layer in memory and **(b)** access to the input is needed to conduct LSH.

With many modern NN architectures reaching billions of parameters in size, requiring resource-constrained devices to conduct LSH analysis over even part of such a large model is infeasible as it requires many linear projections of massive weights. The hope of offloading this memory- and computationally-intensive task to a central host in the distributed setting is equally fruitless. LSH-based pruning cannot be conducted by a central host as it requires access to either local client data or hashed mappings of such data. Both of these violate the fundamental host-client privacy contract, especially in a federated setting [20]. Therefore, in order to maintain privacy, devices are forced to conduct LSH themselves, returning us back to our original drawback in a vicious circle. We raise the following question then:

*Can a resource-constrained device conduct LSH-like pruning of a large dense layer without ever needing to see the entirety of its underlying weight?*

This work makes the following contributions to positively resolve this question:
**(1)** Introduce a novel family of hash functions, PGHash, for the detection of high cosine similarity amongst vectors. PGHash improves upon the efficiency of SimHash by comparing binarized random projections of *folded* vectors. We prove several statistical properties about PGHash, including angle/norm distortion bounds, and that it is an LSH family.
**(2)** Present an algorithmic LSH framework, leveraging our hash functions, which allows for private, personalized, and memory-efficient distributed/federated training of large-scale recommender networks via dynamic pruning.
**(3)** Showcase experimentally that our PGHash-based framework is able to efficiently train large-scale recommender networks. Our approach is competitive against a distributed implementation of SLIDE [8] using full-scale LSH. Furthermore, where entry-magnitude similarity is desired over angular similarity (training over Amazon-670K, for example), we empirically demonstrate that using our DWTA [9] variant of PGHash, PGHash-D, matches the performance of using full-scale DWTA.

## 2 Related work

**LSH Families.** Locality-sensitive hashing families have been used to efficiently solve the approximate nearest neighbors problem [16, 15, 2]. SimHash [6], based on randomized hyperplane projections, is used to estimate cosine similarity. Each SimHash function requires a significant number of random bits if the dimensionality of each target point is large. However, bit reduction using Nisan's pseudorandom generator [23] is often suggested [11, 14]. MinHash [5], a competitor to SimHash [28], measures Jaccard similarity between binary vectors and has been used for document classification. The Winner Take All (WTA) hash [32] compares the ordering of entries by magnitude (corresponding to Kendall-Tau similarity); such comparative reasoning has proven popular in vision applications [25]. However, it was observed that WTA was ineffective at differentiating highly sparse vectors leading to the development of Densified WTA (DWTA) [9]. Since MinHash, WTA, and DWTA are better suited for binary vector comparison, and we require comparison over real-valued vectors, PGHash is founded on SimHash.

**Hash-based Pruning.** One of the earlier proposals of pruning based on input-neuron angular similarity via LSH tables is in [29], where a scheme for asynchronous gradient updates amongst multiple threads training over a batch along with hashed backpropagation are also outlined. These principles are executed to great effect in both the SLIDE [8] and Mongoose [7] frameworks for training extreme-scale recommender networks. Mongoose improves SLIDE by using an adaptive scheduler to determine when to re-run LSH over a layer weight, and by utilizing learnable hash functions. Both works demonstrated that a CPU using LSH-based dynamic dropout could achieve competitive training complexity against a GPU conducting fully-dense training. Reformer [22] uses LSH to reduce the memory complexity of self-attention layers.

**Distributed Recommender Networks.** Several works which prune according to input-neuron angular similarity estimations via LSH utilize multiple workers on a single machine [29, 24, 7, 8]. Federated training of recommender networks is an emerging topic of interest, with particular interest in personalized training [18, 26] malicious clients [30, 36], and wireless unreliability [1]. D-SLIDE [33], which is the federated version of SLIDE, eases local on-device memory and computational requirements by sharding the network across clients. However, in the presence of low client numbers,

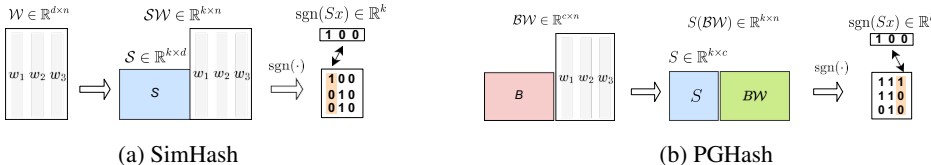

(a) SimHash                        (b) PGHash

Figure 1: **Hash-based dropout.** Two procedures for conducting cosine similarity estimation between full weight matrix $W \in \mathbb{R}^{d \times n}$ (column $w_i$ denotes the weight of neurons $i$) and an input $x \in \mathbb{R}^d$. **(a)** SimHash generates a hash table via left-multiplication by a randomized rectangular Gaussian matrix $S \sim \mathcal{N}(0, I_d)$ onto the *fully-sized* $W$. **(b)** PGHash generates a hash table via left-multiplication by a randomized square Gaussian matrix $S \sim \mathcal{N}(0, I_c)$ onto a base projection $BW \in \mathbb{R}^{c \times n}$ of $W$. In both procedures, weight $w_i$ is selected for activation if its signed hash matches the signed hash of $x$.

the proportion of the model owned per device can still be taxing, whereas our compression is independent of the number of federated agents. In [31], clients query the server for weights based on the results of LSH conducted using server-provided hash functions. We regard this complete server control over the hashing family, and therefore access to hash-encoding of local client data, as non-private and potentially open to honest-but-curious attacks.

## 3 Preliminaries

Let $W = \{w_1, w_2, \ldots, w_n\} \subset \mathbb{R}^d$ be the weights of a dense hidden layer and $x \in \mathbb{R}^d$ be the input. For brevity, we refer to $W \in \mathbb{R}^{d \times n}$ as the *weight* of the layer. In particular, $w_i \in \mathbb{R}^d$ corresponds to the $i^{\text{th}}$ neuron of the layer. We assume that the layer contains $dn$ parameters. Within our work, we perform MIPS, as we select weights that produce large inner products with $x$. Mathematically, we can begin to define this by first letting $p = \max w_i^\top x$ for $1 \le i \le n$. For $0 < \epsilon < 1$ we are interested in selecting $\mathcal{S} \subset W$, such that for $\forall w_i \in S, w_i^\top x > \epsilon p$. The weights of $\mathcal{S}$ will pass through activation while the rest are dropped out, reducing the computational complexity of the forward and backward pass through this layer. As detailed in Section 4 and illustrated in Figure 1, we will determine $\mathcal{S}$ by estimating angles with a projection $BW = \{Bw_i\}_{i=1}^n$, with $B \in \mathbb{R}^{c \times d}$ such that $c << d$.

**Locality-sensitive Hashing (LSH).** LSH [2] is an efficient framework for solving the $\epsilon$-approximate nearest neighbor search (NNS) problem:

> **Definition 1** ($\epsilon$-NNS). *Given a set of points $P = \{p_1, p_2, \ldots, p_n\}$ in a metric space $\mathcal{M}$ and similarity funcion $Sim : \mathcal{M} \times \mathcal{M} \to \mathbb{R}$ over this space, find a point $p \in P$, such that for a query point $q \in X$ and all $p' \in P$, we have that $Sim(p, q) \le (1 + \epsilon)Sim(p', q)$.*

It is important to note that the similarity function $Sim(\cdot)$ need not be a distance metric, but rather any general comparison mapping. Popular choices include Euclidean distance and cosine similarity, the latter of which is the primary focus of this paper. The cosine similarity for $x, y \in \mathbb{R}^d$ is defined as $\cos(x, y) \triangleq x^\top y / (||x||_2 ||y||_2)$. We can frame the MIPS problem described previously as an $\epsilon$-NNS one if we assume that the weights $w_i \in W$ are of unit length. Thus, we are searching for $\epsilon$-nearest neighbors in $W$ of the query $x$ according to cosine similarity.

Consider a family $\mathcal{H}$ containing hash functions of the form $h : \mathcal{M} \to \mathcal{S}$, where $\mathcal{S}$ is a co-domain with significantly lower feature dimensionality than $\mathcal{M}$. We say that $\mathcal{H}$ is locality-sensitive if the hashes of a pair of points $x, y$ in $\mathcal{M}$, computed by an $h \in \mathcal{H}$ (selected uniformly at random), have a higher collision (matching) probability in $S$ the more similar $x$ and $y$ are according to $Sim$. We now formally define this notion following [15].

> **Definition 2** (Locality-sensitive Hashing). *A family $\mathcal{H}$ is called $(S_0, \epsilon S_0, p_1, p_2)$-sensitive if for any two points $x, y \in \mathbb{R}^d$ and $h$ chosen uniformly at random from $\mathcal{H}$ satisfies the following,*
> *1. if $Sim(x, y) \ge S_0 \Rightarrow Pr(h(x) = h(y)) \ge p_1$, 2. if $Sim(x, y) \le \epsilon S_0 \Rightarrow Pr(h(x) = h(y)) \le p_2$.*

For an effective LSH, $p_1 < p_2$ and $\epsilon < 1$ is required. An LSH family allows us to conduct a similarity search over a collection of vectors through a comparison of their hashed mappings. Of course, locality loss is inevitable if $Sim$ is a dimension-lowering projection. Through a mixture of increased precision

(raising the output dimension of $\mathcal{H}$) and repeated trials (running several trials over independently chosen $h$), we may tighten the correspondence between $Sim$ and matches over $\mathcal{H}$, following the spirit of the Johnson-Lindenstrauss Lemma [19].

**SimHash.** A popular LSH algorithm for estimating cosine similarity is SimHash, which uses signed random projections [6] as its hash functions. Specifically, for a collection of vectors $W \subset R^d$, the SimHash family $\mathcal{H}^{Sim}$ consists of hash functions $h_v$, each indexed by a random Gaussian vector $v \sim \mathcal{N}(0, I_n)$, i.e., an $n$-dimensional vector with iid entries drawn from $\mathcal{N}(0,1)$. For $x \in R^n$, we define the hash mapping $h_v(x) := \text{sgn}(v^\top x)$. Here, we modify $\text{sgn}(x)$ to return 1 if $x > 0$, else it returns 0. For Gaussian $v$ chosen uniformly at random and fixed $x, y \in \mathbb{R}^d$, we have $Pr(h_v(x) = h_v(y)) = 1 - \frac{\arccos(\frac{x^\top y}{||x|| \cdot ||y||})}{\pi}$. This hashing scheme was popularized in [12] as part of a randomized approximation algorithm for solving MAX-CUT. Notice that the probability of a hashed pair matching is monotonically increasing with respect to the cosine similarity of $x$ and $y$, satisfying Definition 2. More precisely, if we set $S_0 = \cos(x, y)$ then $\mathcal{H}^{Sim}$ is $\left(S_0, \epsilon S_0, \left(1 - \arccos(\frac{S_0}{\pi})\right), \left(1 - \arccos(\frac{\epsilon S_0}{\pi})\right)\right)$-sensitive [6, 28]. The above discussion considers the sign of a single random projection, but in practice, we will perform multiple projections.

> **Definition 3** (Hash table). *Let $X = \{x_1, \ldots, x_n\} \subset \mathbb{R}^d$ and $V = \{v_1^\top, \ldots, v_k^\top\} \subset \mathcal{N}(0, I_d)$, where $k$ is the **hash length**. Define $h_V : \mathbb{R}^d \to \mathbb{R}^k$ by $[h_V(x)]_i = h_{v_i}(x)$ where $h_{v_i} \in \mathcal{H}^{Sim}$ for $1 \leq i \leq k$. For fixed $V$, the **hash table** $h_V(X) \in \mathbb{R}^{k \times n}$ is a binary matrix with columns $h_V(x_j)$ for $1 \leq j \leq n$.*

Following the notation above, we may estimate the similarity between an input $q \in \mathbb{R}$ and a collection of vectors $X \subset \mathbb{R}^d$ by measuring the Hamming distances (or exact sign matches) between $h_V(q)$ and columns of $h_V(X)$. SimHash is now more discriminatory, as $\mathcal{H}^{Sim}$ can separate $R^d$ into $2^k$ buckets corresponding to all possible length $k$ binary vectors (which we refer to as **hash codes**). Finally, counting the frequency of exact matches or computing the average Hamming distance over several independently generated hash tables further improves our estimation of closeness. Implementations of the well-known SLIDE framework [8, 24], which utilize SimHash for LSH-based weight pruning, require upwards of 50 tables.

**DWTA.** Another popular similarity metric is to measure how often high-magnitude entries between two vectors occur at the exact same positions. The densified winner-take-all (DWTA) LSH family [10] estimates this similarity by uniformly drawing $k$ random coordinates over $W$ and recording the position of the highest-magnitude entry. Similar to SimHash, this process is repeated several times, and vectors with the highest frequency of matches are expected to have similar magnitude ordering. This type of comparative reasoning is useful for computer vision applications [37].

## 4 PGHash

In this section, we develop a family of hash functions $\mathcal{H}^{PG}$ which allow for memory-efficient serial generation of hash tables using a single dimensionally-reduced sketch of $W$. This is in contrast to traditional LSH frameworks, which produce hash tables via randomized projections over the entirety of $W$. We first present an overview of the Periodic Gaussian Hash (PGHash) followed by its algorithm for distributed settings, an exploration of several statistical properties regarding the local sensitivity of $\mathcal{H}^{PG}$.

**PGHash Motivation.** As detailed in Section 3, our goal is to efficiently estimate cosine similarity between an input to a layer $x \in \mathbb{R}^d$ and the columns of a large weight matrix $W \in \mathbb{R}^{d \times n}$. SimHash performs hash table generation by first multiplying a matrix of uniformly drawn Gaussian hyperplanes $T_V \in \mathbb{R}^{c \times d}$ with $W$. The full hash table is computed as $h_V(W) = \text{sgn}(T_V W)$. Then, the $j^{\text{th}}$ neuron is activated if $\text{sgn}(T_V x) = h_V(w_j)$ for a layer input $x$.

One can immediately notice that generation of a new hash table $h_{\tilde{V}}(W)$ requires both **(i)** computation of $T_{\tilde{V}} W$ which requires access to the fully-sized weights and **(ii)** the storage of $T_{\tilde{V}}$ to compute $\text{sgn}(T_{\tilde{V}} x')$ for further inputs $x'$. *This is problematic for a memory-constrained device, as it would need to maintain both $W$ and $T_{\tilde{V}}$ to generate further tables and perform dynamic pruning.* To solve this issue, we introduce a family of hash functions generated from a single projection $BW$ of $W$.

## 4.1 PGHash theory

**Definition 4** (Periodic Gaussian Hash). *Assume **sketch dimension** $c << d$ divides $d$ for simplicity. Let $B = [I_c|I_c|\cdots|I_c] \in \mathbb{R}^{c\times d}$, where $I_c$ is the $c \times c$ identity matrix and $|$ denotes $\frac{d}{c}$ concatenations. Let $S \in \mathbb{R}^{k\times c}$ be a random Gaussian matrix with iid entries drawn from $\mathcal{N}(0,1)$. We may define a **Periodic Gaussian Hash (PGHash)** function $h_S^{PG} : \mathbb{R}^d \to \mathbb{R}^k$ by $[h_S^{PG}(x)]_i = [sgn(SBx)]_i$ for $1 \le i \le k$. We denote the family of all such hash functions as $\mathcal{H}^{PG}(c,d)$.*

We use the term "periodic" to describe the hash functions described in Definition 4, since unlike SimHash which projects a point via a fully random Gaussian vector as in SimHash, our projection is accomplished using a repeating concatenation of a length $c$ Gaussian vector. Furthermore, for $h_S^{PG} \subset H^{PG}(c,d)$, the matrix representation $SB$ is a tiling of a single Gaussian matrix. Notice that we may easily extend the notion of a PGHash of one vector to an entire hash table over multiple vectors following Definition 3. In this manner, we may generate a sequence of hash tables $\{h_{S_i}^{PG}(W)\}_{i=1}^{\tau}$ over a weight matrix $W$ simply by drawing random Gaussian matrices $S_i \in \mathbb{R}^{k\times c}$ for $1 \le i \le \tau$ (where $k$ is the hash length) and computing $\text{sgn}(S_iBW)$.

**Extension to DWTA (PGHash-D) Remark.** When DWTA (described in Section 3) is preferred over SimHash for similarity estimation, we may modify Definition 4 as follows: $B = D_{\mathbb{1}}P$ where $P$ is a $d \times d$ random permutation matrix, and $D_{\mathbb{1}}$ is a $c \times d$ rectangular diagonal matrix with $D_{ii} = 1$ for $1 \le i \le c$. We denote our hash functions as $h_S^{PG-D} : \mathbb{R}^d \to \mathbb{R}^k$ by $h_S^{PG-D}(x) = \max_i[SBx]_i$ for $1 \le i \le k$, with $k \le c$, where $S$ is now a rectangular permutation matrix which selects $k$ rows of $Bx$ at random. We refer to this scheme as PGHash-D, whereas PGHash refers to Definition 4.

**Local Memory Complexity Remark.** When generating a new table using PGHash, a device maintains $S$ and needs access to just $BW$, which costs $\mathcal{O}(kc)$ and $\mathcal{O}(cn)$ space complexity respectively. This is much smaller than the $\mathcal{O}(kd)$ and $\mathcal{O}(dn)$ local memory requirements of SimHash.

**Sensitivity of $\mathcal{H}^{PG}$.** In this section, we will explore the sensitivity of $\mathcal{H}^{PG}(c,d)$.

**Definition 5.** *Let $x \in \mathbb{R}^d$ and $c \in \mathbb{R}$ such that $c|d$. Define the $(d,c)$-**folding** $x_c \in \mathbb{R}^c$ of $x$ as $[x_c]_i = \sum_{j=1}^{\frac{d}{c}} [x]_{i+j\cdot\frac{d}{c}}$. Equivalently, $x_c = Bx$, with $B$ as specified in Definition 4.*

**Theorem 1.** *Let $x,y \in R^d$. Define the following similarity function $Sim_c^d(x,y) \triangleq \cos(x_c, y_c)$, where $x_c, y_c$ are $(d,c)$-foldings of $x, y$. $\mathcal{H}^{PG}(c,d)$ is an LSH family with respect to $Sim_c^d$.*

*Proof.* Let $h_v \in \mathcal{H}^{PG}$. This means that for a randomly chosen $v' \sim \mathcal{N}(0, I_{\frac{d}{c}})$, $v$ is a $c$-times concatenation of $v$. We see that $\text{sgn}(v^\top x) = \text{sgn}\left(\frac{v^\top x}{||v||\cdot||x||}\right) = \text{sgn}\left(\sqrt{\frac{c}{d}}\frac{||x_c||}{||x||}\frac{v'^\top x_c}{||v'||||x_c||}\right)$. Since sgn is unaffected by the positive multiplicative factors, we conclude that $\text{sgn}(v^\top x) = \text{sgn}(v'^\top x_c)$. Through symmetric argument, we find $\text{sgn}(v^\top y) = \text{sgn}(v'^\top y_c)$. Since $v' \sim \mathcal{N}(0, I_c)$, comparing the sign of $v^\top x$ to $v^\top y$ is equivalent to a standard SimHash over $x_c$ and $y_c$, i.e., estimation of $\cos(x_c, y_c)$. $\square$

**Corollary 1.** *Let $x,y \in \mathbb{R}^d$, then $\mathcal{H}^{PG}(c,d)$ is $\left(S_c, \epsilon S_c, \left(1 - \arccos(\frac{S_c}{\pi})\right), \left(1 - \arccos(\frac{\epsilon S_c}{\pi})\right)\right)$-sensitive where $S_c = \cos(x_c, y_c)$,.*

*Proof.* This follows directly from the well-known sensitivity of SimHash [6]. $\square$

We see that $\mathcal{H}^{PG}$ is LSH with respect to the angle between $(d,c)$-foldings of vectors. The use of periodic Gaussian vectors restricts the degrees of freedom (from $d$ to $d/c$) of our projections. However, the usage of pseudo-random and/or non-iid hash tables has been observed to perform well in certain regimes[3, 35]. *Although $\mathcal{H}^{PG}(c,d)$ is LSH, is $\cos(x_c, y_c)$ necessarily an acceptable proxy for $\cos(x,y)$, in particular, for high angular similarity?* Heuristically, yes, for highly-cosine similar vectors: assuming $x$ and $y$ are both unit (since scaling does not affect angle) then we have that $||x-y||^2 = 2 - 2\cos(x,y)$. If $\ell_2$ similarity between $x$ and $y$ is already high, then the $\ell_2$ similarity

of their (normalized) $(d, c)$-foldings will also be high, and thus their cosine similarity as well. We now provide a characterization of the angle distortion of a $(d, c)$-folding.

> **Theorem 2.** *Let $x, y \in \mathbb{S}^{d-1}$. Assume that neither $x$ nor $y$ vanish under multiplication by $B$ and define the set $S_{x,y} := \{v \in \text{span}(x, y) : ||v|| = 1\}$. We denote the following quantities: $\lambda = d/c$, $\theta := \frac{1}{2}\arccos(\cos(x, y))$, $\alpha = \inf\{c > 0 \mid |||Bv|| \geq c, \forall v \in S_{x,y}\}$, and $\beta = \alpha/\lambda$. Assume $\alpha > 0$. Then $\cos(x_c, y_c)$ lives between $\frac{1-\beta^2 \tan^2 \theta}{1+\beta^2 \tan^2 \theta}$ and $-\frac{\tan^2 \theta^2 - \beta^2}{\tan^2 \theta^2 + \beta^2}$.*

*Proof sketch.* Consider the unit circle $S_{x,y}$ contained in $\text{span}(x, y)$ (Let us assume $x$ and $y$ are unit, WLOG). The linear distortion $BS_{x,y}$ is an ellipse containing $x_c = Bx$ and $y_c = By$. The length of the axes of this ellipse is determined by the eigenvalues of $B^\top B$. The bounds follow from further trigonometric arguments, by considering when the axes of $BS_{x,y}$ are maximally stretched and shrunk respectively. These distortions are strongly related to $\lambda$ and $\beta$.

We can see that as $\cos(x, y) \rightarrow 0$ we have $\cos(x_c, y_c) \rightarrow 0$. It is natural to consider the distribution of $\alpha$ in Theorem 2 as how extreme shrinking by $B$ (the folding matrix) can greatly distort the angle. We can characterize this statistical distribution exactly.

**Proposition 1.** *Let $u \in S^{d-1}$, be drawn uniformly at random. Then $||Bu||^2 \sim \text{Beta}(\frac{c}{2}, \frac{d-c}{2}, 0, \frac{d}{c})$, the four parameter Beta distribution with pdf $f(x) = \frac{(2x/c)^{c/2-1}(1-2x/c)^{(d-c)/2-1}}{(d/c)\beta(c/2,(d-c)/2}$ and $\mathbb{E}||Bu||^2 = 1$.*

We defer proof of Proposition 1 to the Appendix D. Since the folded magnitude $||Bu||^2$ is unit in expectation, the distortion term $\alpha^2/\lambda^2$ in Theorem 2 will often be close to 1, greatly tightening the angle distortion bounds.

### 4.2 PGHash algorithm

Below, we detail our protocol for deploying PGHash in a centralized distributed setting (presented algorithmically in Algorithm 1). Over a network of $N$ devices, the central host identifies a target layer $P$ whose weight $W_t^P \in \mathbb{R}^{d \times n}$ (at

---

**Algorithm 1** `Distributed PGHash`

---

**Require:** weights $\{W_0^\ell\}_{\ell=1}^L$, objective $\mathcal{F}$, $N$ devices, target layer $P$, $T$ iterations, folding matrix $B$, $\tau$ hash tables, hash length $k$, sketch dim $c$, comp. ratio $CR$.
1: **Server Executes:**
2: **for** $t = 1, 2, \ldots, T$ **do**
3:     Compile pre-target weights $W_A = \{W_t^\ell\}_{\ell=1}^{P-1}$
4:     Compile post-target weights $W_B = \{W_t^\ell\}_{\ell=P+1}^{L}$
5:     **for** each device $i$ **in parallel do**
6:        $\Theta_i \leftarrow$ **DeviceLSH**(i, $W_A$, $BW_t^P$, $CR$, $\tau$, $k$, $c$)
7:        $W_{t+1,i} \leftarrow$ **DeviceUpdate**(i, $W_t^P(\Theta_i) \bigcup W_B$)
8:     **end for**
9:     $W_{t+1} \leftarrow$ Average the active weights $W_{t+1,i}$ across all devices
10: **end for**
11: **return** $\{W_T^\ell\}_{\ell=1}^L$

12: **DeviceLSH**($i, W_A, BW, CR, \tau, k, c$):
13: Sample input $\xi$ from local data distribution $\mathcal{D}_i$
14: $\xi^{P-1} = \mathcal{F}(W_A, \xi)$ $\underline{W_A \text{ is stored locally}}$
15: $\Theta_i \leftarrow$ `PGHash`($\xi^{P-1}, BW, CR, \tau, k, c$)
16: **return** $\Theta_i$

17: **DeviceUpdate**($i, W_B$):
18: $W \leftarrow W_A \bigcup W_B$
19: **return** $W - \eta \nabla_W \mathcal{F}(W; \xi)$

---

iteration $t$) is too expensive for memory-constrained devices to fully train or host in local memory. Neurons (columns of $W_t^P$) are pruned by devices according to their estimated cosine similarity to the output $x^{P-1}$ of the previous layer.

The central host begins each round of distributed training $t$ by sending each device **(1)** all weights $\{W_t^\ell\}_{\ell=1}^{P-1}$ required to generate the input $x^{P-1}$ and **(2)** the compressed target layer $BW_t^P$. Using these weights, each device conducts PGHash analysis (via Algorithm 2) using its current batch of local data to determine its activated neurons. The central host sends each device $i$ their set of activated neurons $W_t^P(\Theta_i)$, and each device performs a standard gradient update on their new model $\{W_t^\ell\}_{\ell=1}^{P-1} \bigcup W_t^P(\Theta_i) \bigcup \{W_t^\ell\}_{\ell=P+1}^L$. Finally, the central host receives updated models from each device and averages only the weights that are activated during training.

The on-device PGHash analysis (Algorithm 2) consists of first running a forward pass up to the target layer to generate the input $x^{P-1}$. Devices generate personal hash tables by performing left-multiplication of $BW_t^P$ and $x^{P-1}$ by a random Gaussian matrix $S \sim \mathcal{N}(0, I_c)$, as described in Section 3. The number of tables $\tau$ is specified by the user. A neuron $j$ is marked as active if the input

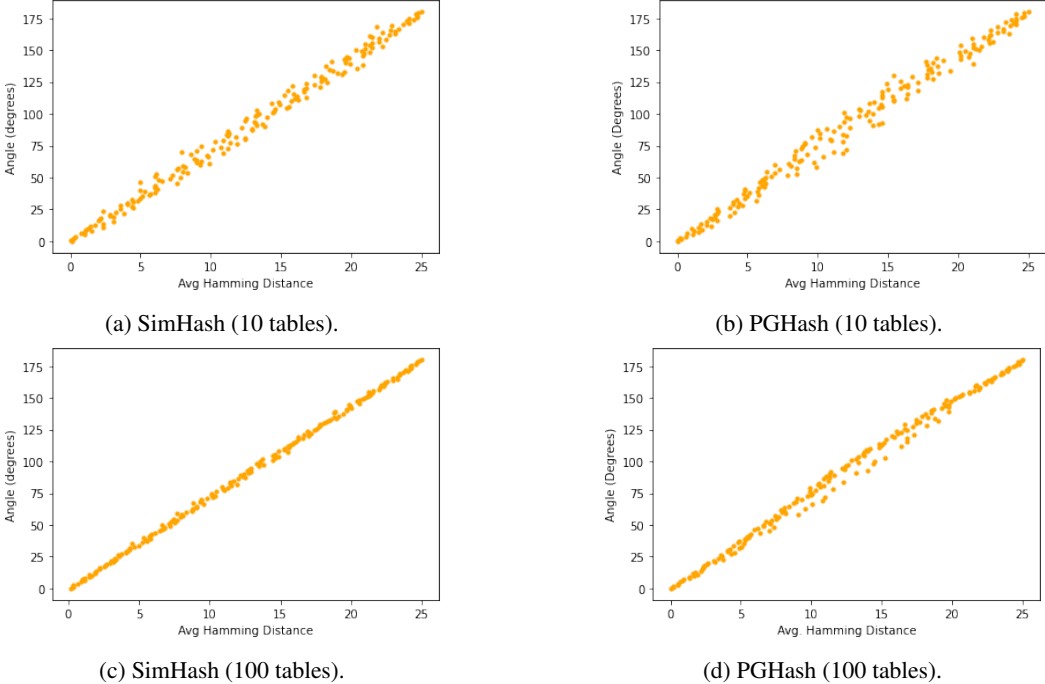

(a) SimHash (10 tables).      (b) PGHash (10 tables).

(c) SimHash (100 tables).      (d) PGHash (100 tables).

Figure 2: **Correlation between angle and Hamming distance.** We plot the average Hamming distance (x-axis) between a PG/SimHash hashed fixed vector $x$ and collection of vectors $\mathcal{V}$ versus their true angles (y-axis). Vectors are unit, length 100, and hashed down to dimension $k = 25$ binary vectors according to $\tau = 10$ or 100 hash tables. PGHash has a sketch dimension of $c = 25$. Both PGHash and SimHash show strong correlation between Hamming distance and angular similarity.

hash code $\text{sgn}(Sx^{P-1})$ is identical to $j$-th weight's hash code $\text{sgn}(SB[W_t^P]_{:,j})$. In Appendix A.2, we detail how Hamming distance can also be used for neuron selection.

**Computational Complexity Remark.** Through the use of dynamic pruning, PGHash significantly reduces both the forward and backward training computational complexities. PGHash activates *at most* $CR \cdot n$ neurons per sample as opposed to $n$ for full training. In practice, PGHash activates only a fraction of the $CR \cdot n$ neurons (as shown in Figure 6a). Therefore, the number of floating point operations within forward and backward training is dramatically reduced.

**Communication Complexity Remark.** By reducing the size of the model needed in local memory and subsequently requesting a pruned version of the architecture we improve communication efficiency. For a fixed number of rounds $T$ and target weight size $dn$, the total communication complexity, with respect to this data structure, is $\mathcal{O}(T \cdot CR \cdot dn)$, which is significantly fewer bits than the vanilla $\mathcal{O}(Tdn)$ communication cost of vanilla federated training. In Section 5, we show that PGHash achieves near state-of-the-art results with only $CR = 0.1$ (10% of a massive weight matrix).

---

**Algorithm 2** PGHash

**Require:** batched input $X \in \mathbb{R}^{d \times M}$, projected weight $BW \in \mathbb{R}^{c \times n}$, compression rate $CR$, $\tau$ hash tables, hash length $k$, sketch dim $c$.
1: Set $\Theta = [\ ]$
2: **for** $i = 1, 2, \ldots, \tau$ **do**
3:      Draw random Gaussian $S \in \mathbb{R}^{k \times c}$
4:      **for** each sample $x$ in $X$ **do**
5:          **for** $j = 1, 2, \ldots, n$ **do**
6:              **if** $\text{sgn}(SBx) = [\text{sgn}(SBW)]_{:,j}$ **then**
7:                 $\Theta.\text{add}(j)$
8:              **end if**
9:          **end for**
10:      **end for**
11:      **if** $|\Theta| > \lfloor CR \cdot n \rfloor$ **then**
12:          Randomly remove selected neurons from table-$i$ until $|\Theta| = \lfloor CR \cdot n \rfloor$
13:          **return** $\Theta$
14:      **end if**
15: **end for**
16: **return** $\Theta$

---

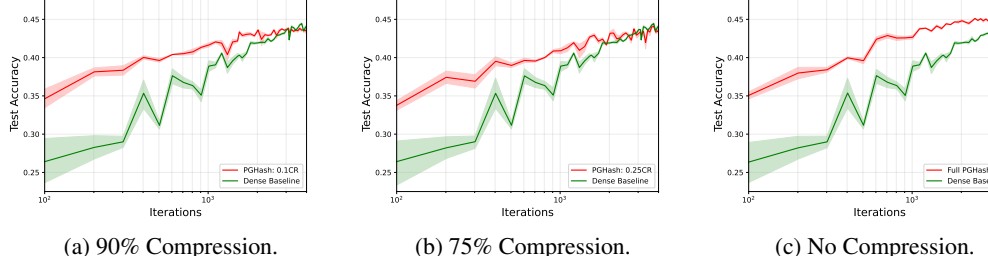

| (a) 90% Compression. | (b) 75% Compression. | (c) No Compression. |

Figure 3: **Compressed PGHash.** We record model accuracy of a large recommendation system on an extreme classification task (Delicious-200K) using PGHash for varying compression rates ($CR$). Compressed PGHash, even at 90% compression, is competitive with full training (*without even including effects of sparsity-induced neuronal drop out*). Hyperparameters are in Appendix A.1.

## 5  Experiments

In this section, we **(1)** gauge the sensitivity of PGHash and **(2)** analyze the performance of PGHash and our own DWTA variant (PGHash-D) in training large-scale recommender networks. PGHash and PGHash-D require only 6.25% ($c = 8$) of the final layer sent by the server to perform on-device LSH in our experiments. In PGHash, devices receive the compressed matrix $BW \in \mathbb{R}^{c \times n}$ via the procedure outlined in Section 4. In PGHash-D, devices receive $c$ out of $d$ randomly selected coordinates for all $n$ neurons in the final layer weight. Using $k$ of the $c$ coordinates (ensuring privacy since the server is unaware of the coordinates used for LSH), PGHash-D selects neurons that, within the $k$ coordinates, share the same index of highest-magnitude entry between the input and weight. We employ PGHash for Delicious-200K and PGHash-D for Amazon-670K and WikiLSHTC-325K.

**PGHash Sensitivity Analysis.**  Our first experiment measures the ability of $\mathcal{H}^{PG}(c, d)$ to estimate cosine similarity. We produce a fixed unit vector $x \in \mathbb{R}^{100}$ and set of 180 vectors $\{v_i\}_{i=1}^{180}$ of the same dimension. Both the Gaussian vector $x$ and collection of vectors $V$ are fed through varying numbers of SimHash and PGHash tables. We produce a scatter plot measuring the correlation between the angle and average Hamming distance. PGHash, as seen in Figure 2, is an effective estimator of cosine similarity. We observe that PGHash, like SimHash, successfully produces low average Hamming distances for vectors that are indeed close in angle. This provides evidence that selecting neurons with exact hash code matches (vanilla sampling) is effective for choosing neurons that are close in angle to the input vector. Finally, we find increasing the number of hash tables helps reduce variance.

**Large-Scale Recommender Network Training.**  Our second experiment tests how well PGHash(-D) can train large-scale recommender networks. We train these networks efficiently by utilizing dynamic neuronal dropout as done in [8]. We use three extreme multi-label datasets for training recommender networks: Delicious-200K, Amazon-670K, and WikiLSHTC-325K. These datasets come from the Extreme Classification Repository [4]. The dimensionality of these datasets is large: 782,585/205,443 (Delicious-200K), 135,909/670,091 (Amazon-670K), and 1,617,899/325,056 (WikiLSHTC-325K) features/labels. Due to space, Wiki results are found in Appendix A.3.

The feature and label sets of these datasets are extremely sparse. Akin to [8, 7, 34], we train a recommender network using a fully-connected neural network with a single hidden layer of size 128. Therefore, for Amazon-670K, our two dense layers have weight matrices of size $(135,909 \times 128)$ and $(128 \times 670,091)$. The final layer weights output logits for label prediction, and we use PGHash(-D) to prune its size to improve computational efficiency during training.

Unlike [8, 7, 34], PGHash(-D) can be deployed in a federated setting. Within our experiments, we show the efficacy of PGHash for both single- and multi-device settings. Training in the federated setting (following the protocols of Algorithm 1) allows each device to rapidly train portions of the entire neural network in tandem. We partition data evenly (in an IID manner) amongst devices. Finally, we train our neural network using TensorFlow. We use the Adam [21] optimizer with an initial learning rate of 1e-4. A detailed list of the hyper-parameters we use in our experiments can be found in Appendix A.1. Accuracy in our figures refers to the $P@1$ metric, which measures whether the predicted label with the highest probability is within the true list of labels. These experiments are run on a cloud cluster using Intel Xeon Silver 4216 processors with 128GB of total memory.

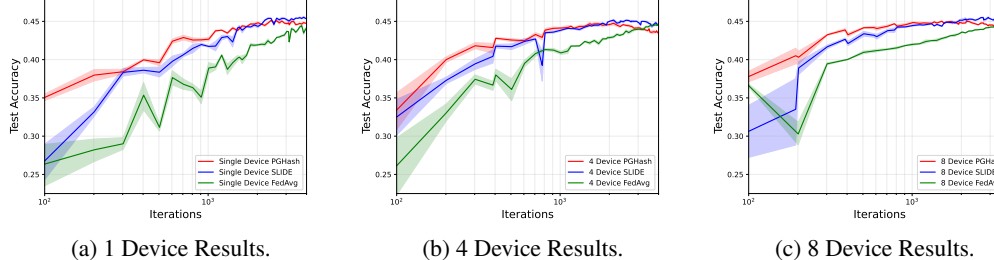

| (a) 1 Device Results. | (b) 4 Device Results. | (c) 8 Device Results. |

Figure 4: **Federated Delicious-200K PGHash.** We record model accuracy of a large recommendation system on an extreme classification task (Delicious-200K) trained in a federated setting. PGHash achieves competitive accuracies compared with Federated SLIDE and FedAvg. In fact, PGHash converges quicker to near-optimal accuracy. Hyperparameters are in Appendix A.1.

**Sampling Strategy.** One important aspect of training is how we select activated neurons for each sample through LSH. Like [8], we utilize vanilla sampling. In our vanilla sampling protocol, a total of $CR \cdot n$ neurons are selected across the entire sampled batch of data. As detailed in Section 4 and Algorithm 2, a neuron is selected when its hash code exactly matches the hash code of the input. We retrieve neurons until either $CR \cdot n$ are selected or all $\tau$ tables have been looked up.

**Compression Efficacy.** We begin by analyzing how PGHash performs when varying the compression rate $CR$. Figure 3 showcases how PGHash performs for compression rates of 75% and 90% as well as no compression. Interestingly, PGHash reaches near-optimal accuracy even when compressed. This shows the effectiveness of PGHash at accurately selecting fruitful active neurons given a batch of data. The difference between the convergence of PGHash for varying compression rates lies within the volatility of training. As expected, PGHash experiences more volatile training (Figures 3a and 3b) when undergoing compression as compared to non-compressed training (Figure 3c).

**Distributed Efficacy.** In Figures 4 and 5, we analyze how well PGHash(-D) performs in a federated setting. We compare PGHash(-D) to a federated version of SLIDE [8] that we implemented (using respectively, a full SimHash or DWTA), as well as fully-dense Federated Averaging (FedAvg) for Delicious-200K. One can immediately see in Figures 4 and 5 that PGHash(-D) performs identically to, or better than, Federated SLIDE. In fact, for Delicious-200K, PGHash and Federated SLIDE outperform the dense baseline (FedAvg). In Appendix A.3 we detail the difficulties of PGHash-D and SLIDE in matching the dense baseline as well as the failure of SimHash to achieve performance akin to DWTA for Amazon-670K.

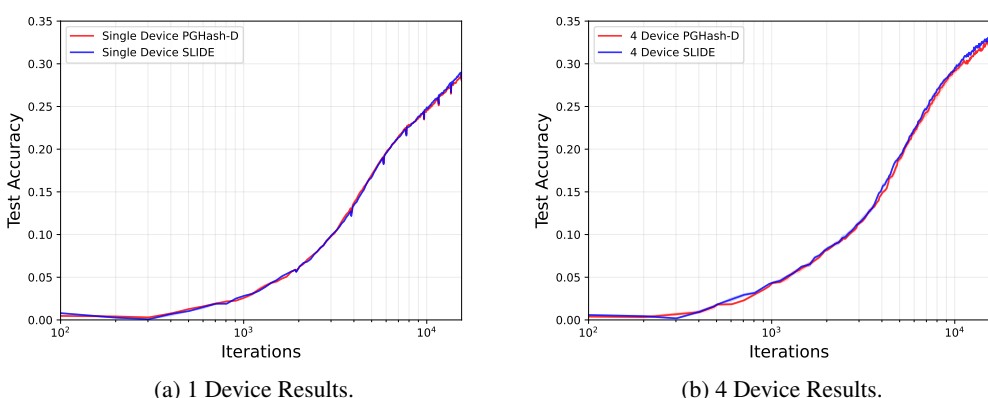

| (a) 1 Device Results. | (b) 4 Device Results. |

Figure 5: **Federated Amazon-670K PGHash-D.** We record model accuracy of a large recommendation system on Amazon-670K trained in a federated setting. PGHash-D matches the convergence of Federated SLIDE without requiring LSH to be performed by the central server.

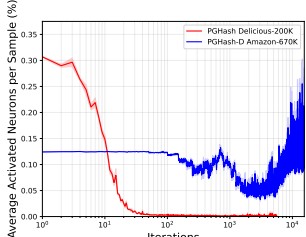 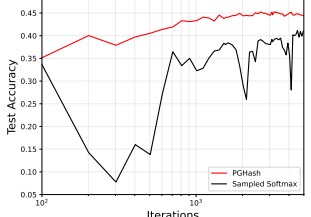 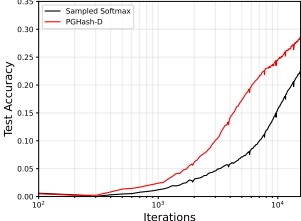

(a) Average Activated Neurons.   (b) Sampled Softmax (Delicious).   (c) Sampled Softmax (Amazon).

Figure 6: **PGHash(-D) Computational Efficiency.** In Figure **6a**, we showcase that PGHash(-D) activates only a fraction of the total final layer neurons *even without compression*. Through this induced sparsity, PGHash(-D) greatly reduces the computational complexity of forward and backward training compared to full training. In Figures **6b** and **6c**, we compare PGHash(-D) with the Sampled Softmax heuristic [17] (randomly sampling 10% of the total neurons) for efficiently training recommender systems. PGHash(-D) outperforms the Sampled Softmax baseline, as it selects a better set of activated neurons via LSH to more efficiently train the recommender system.

PGHash(-D) and Federated SLIDE smartly train portions of the network related to each batch of local device data, via LSH, in order to make up for the lack of a full output layer. However, unlike Federated SLIDE, PGHash(-D) can perform on-device LSH *using as little as 6.25% of the full weight $W$ ($c = 8$)* for both Delicious-200K and Amazon-670K experiments. Furthermore, for Delicious-200K, PGHash generates a dense Gaussian that is only 6.25% ($c = 8$) the size of that for Federated SLIDE. In summary, PGHash(-D) attains similar performance to Federated SLIDE while storing less than a tenth of the parameters.

**Induced Sparsity.**    PGHash(-D) induces a large amount of sparsity through its LSH process. This is especially prevalent in large-scale recommender networks, where the number of labels for each data point is a minuscule fraction of the total output layer size (*e.g.* Delicious-200K has on average only 75.54 labels per point). PGHash(-D) performs well at identifying this small subset of neurons as training progresses. As one can see in Figure 6a, even when PGHash is allowed to select all possible neurons (*i.e.,* no compression $CR = 1$), it still manages to *select fewer than 1% of the total neurons after only 50 iterations of training over Delicious-200K*. For Amazon-670K, PGHash-D requires less than 30% of the total neurons for the majority of training. Therefore, PGHash(-D) greatly increases sparsity within the NN, improving the computational efficiency of the algorithm by reducing the number of floating point operations required in the forward and backward training.

## 6   Conclusion

In this work, we present a new hashing family, PGHash, which enables the generation of multiple LSH hash tables using a single base projection of a massive target weight. These hash tables can be used to dynamically select neurons that are similar to the layer input. This alleviates memory, communication, and privacy costs associated with conventional LSH-training approaches. As a proof of concept, we demonstrate that (i) the PGHash family is effective at mimicking SimHash and (ii) our framework is competitive against other, memory-inefficient, LSH-based federated training baselines of large-scale recommender networks. For future work, we intend to explore how multi-layer PGHash pruning affects model performance and incorporate learnable hashes as in the Mongoose [7] pipeline.

**Limitations.**    Our theory indicates that PGHash is useful for detecting high angular similarity, but could prove unreliable for differentiating between intermediately dissimilar vectors. Additionally, LSH-based pruning has only shown success on large classification layers or attention layers in transformers [22]. When considering broader impacts, large-scale recommender networks, and any subsequent improvements to their design, can be used for strategically negative advertising purposes.

## Acknowledgements

Rabbani, Bornstein, and Huang are supported by the National Science Foundation NSF-IIS-FAI program, DOD-ONR-Office of Naval Research, DOD Air Force Office of Scientific Research, DOD-DARPA-Defense Advanced Research Projects Agency Guaranteeing AI Robustness against Deception (GARD), Adobe, Capital One, and JP Morgan faculty fellowships. Rabbani was supported in part by NSF award DGE-1632976.

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

# Large-Scale Distributed Learning via Private On-Device LSH Appendix

## A Experiment details

In this section, we provide deeper background into how our experiments were run as well as some additional results and observations. We first detail the hyper-parameters we used in order to reproduce our results. Then, we provide additional comments and details into our sampling approach. Finally, we describe some of the interesting observations we encountered while solving the Amazon-670K and Wiki-325K recommender system problems.

### A.1 Experiment hyper-parameters

Below, we detail the hyper-parameters we used when running our federated experiments.

Table 1: **Hyper-parameters for Federated Experiments (PGHash and Federated SLIDE).**

| Dataset | Algorithm | Hash Type | LR | Batch Size | Steps per LSH | $k$ | $c$ | Tables | $CR$ |
|---|---|---|---|---|---|---|---|---|---|
| Delicious-200K | PGHash | PGHash | 1e-4 | 128 | 1 | 8 | 8 | 50 | 1 |
| Delicious-200K | SLIDE | SimHash | 1e-4 | 128 | 1 | 8 | N/A | 50 | 1 |
| Amazon-670K | PGHash | PGHash-D | 1e-4 | 256 | 50 | 8 | 8 | 50 | 1 |
| Amazon-670K | SLIDE | DWTA | 1e-4 | 256 | 50 | 8 | N/A | 50 | 1 |
| Wiki-325K | PGHash | PGHash-D | 1e-4 | 256 | 50 | 5 | 16 | 50 | 1 |
| Wiki-325K | SLIDE | DWTA | 1e-4 | 256 | 50 | 5 | N/A | 50 | 1 |

What one can immediately see from Table 1, is that we use a Densified Winner Take All (DWTA) variant of PGHash for the larger output datasets Amazon-670K and Wiki-325K. As experienced in [8, 7, 24], SimHash fails to perform well on these larger datasets. We surmise that SimHash fails due in part to its inability to select a large enough number of neurons per sample (we observed this dearth of activated neurons empirically). Reducing the hash length $k$ does increase the number of neurons selected, however this decreases the accuracy. Therefore, DWTA is used because it utilizes more neurons per sample on these larger problems and also still achieves good accuracy.

Table 2: **Hyper-parameters for Compression Experiments (PGHash).**

| Dataset | Algorithm | Hash Type | LR | Batch Size | Steps per LSH | $k$ | $c$ | Tables | $CR$ |
|---|---|---|---|---|---|---|---|---|---|
| Delicious-200K | PGHash | PGHash | 1e-4 | 128 | 1 | 8 | 8 | 50 | 0.1/0.25/1 |

As a quick note, we record test accuracy every so often (around 100 iterations for Delicious-200K and Amazon-670K). Similar to [8], to reduce the test accuracy computations (as the test sets are very large) we compute the test accuracy of 30 randomly sampled large batches of test data.

### A.2 Neuron sampling

**Speed of Neuron Sampling.** In Table 3 we display the time it takes to perform LSH for PGHash given a set number of tables. These times were collected locally during training. The entries in Table 3 denote the time it takes to compute hashing of the final layer weights $w_i$ and each sample $x$ in batch $M$ as well as vanilla-style matching (neuron selection) for each sample.

Table 3: **Average LSH time for PGHash over a range of tables.** We compute the average $\mu$ time (and standard deviation $\sigma$) it takes for PGHash to perform *vanilla sampling* (exact matches) between the hash codes of sample $x$ and each weight $w_i$ in the final dense layer. Times are sampled for PGHash on Delicious-200k for batch size $M = 128$, $k = 9$, and $c = 8$ for one device.

| Method | 1 table (seconds) | 50 tables (seconds) | 100 tables (seconds) |
|---|---|---|---|
| PGHash | $\mu = 0.0807, \sigma = 0.0076$ | $\mu = 3.1113, \sigma = 0.0555$ | $\mu = 6.2091, \sigma = 0.1642$ |
| SLIDE | $\mu = 0.0825, \sigma = 0.0099$ | $\mu = 3.2443, \sigma = 0.1671$ | $\mu = 6.2944, \sigma = 0.0689$ |

We find in Table 3 that PGHash achieves near sub-linear speed with respect to the number of tables $\tau$ and slightly outperforms SLIDE. PGHash edges out SLIDE due to the smaller matrix multiplication cost, as PGHash utilizes a smaller random Gaussian matrix (size $c \times c$). The speed-up over SLIDE will

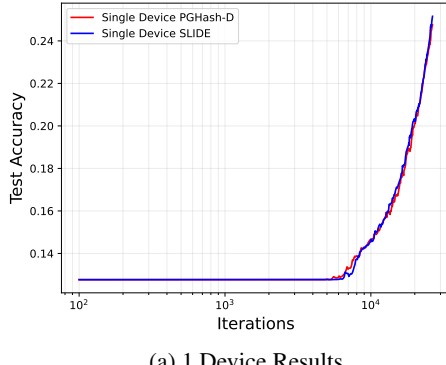
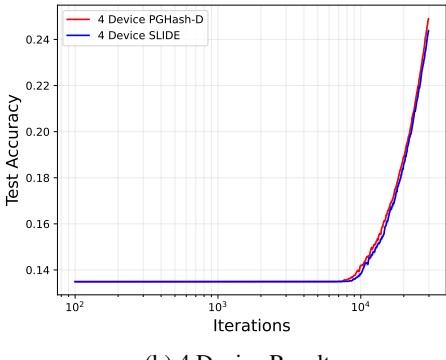

(a) 1 Device Results.  (b) 4 Device Results.

Figure 7: **Wiki-325K PGHash-D.** We record model accuracy of a large recommendation network on Wiki-325K trained in a federated setting. PGHash-D matches or outperforms the convergence of SLIDE without requiring LSH to be performed by the central server.

become more significant when the input layer is larger (as $d = 128$ in our experiments). Therefore, PGHash obtains superior sampling performance to SLIDE.

**Hamming Distance Sampling.**  An alternative method to vanilla sampling is to instead select final layer weights (neurons) $w_i$ which have a small Hamming distance relative to a given sample $x$. As a refresher, the Hamming distance simply computes the number of non-matching entries between two binary codes (strings). If two binary codes match exactly, then the Hamming distance is zero. In this sampling routine, either **(i)** the top-k weights $w_i$ with the smallest Hamming distance to sample $x$ are selected to be activated or **(ii)** all weights $w_i$ with a Hamming distance of $\beta$ or smaller to sample $x$ are selected to be activated. Interestingly, the vanilla-sampling approach we use in our work is equivalent to using $\beta = 0$ in **(ii)**.

In either of the scenarios listed above, hash codes for $w_i$ and $x$ are computed as done in PGHash(-D). From there, however, the hash code for $x$ is compared to the hash codes for all final layer weights in order to compute the Hamming distance for each $w_i$. The process of computing $n$ Hamming distances for each sample $x$ is very expensive (much harder than just finding exact matches). That is why our work, as well as [8, 7], use vanilla sampling instead of other methods.

### A.3  Amazon-670K and Wiki-325K experiment analysis

**Sub-par SimHash Performance.**  SimHash is known to perform worse than DWTA on Amazon-670K and Wiki-325K. Utilizing SimHash for these experiments is unfair as it is shown by [8, 7], for example, that DWTA achieves much higher performance on Amazon-670K. For this reason, DWTA is the chosen hash function in [8] for Amazon-670K experiments. To verify this observation, we performed experiments on Amazon-670K with PGHash (not PGHash-D) and SLIDE (with a SimHash hash function). Table 4 displays the SimHash approach for Amazon-670K.

Table 4: **PGHash and SLIDE performance on Amazon-670K using SimHash.** Accuracy across the first 5,000 iterations for a single device. Batch size $M = 1024$, $k = 8$, and $c = 8$.

| Iteration | SLIDE | PGHash |
|-----------|-------|--------|
| 1,000 | 10.82% | 10.04% |
| 2,000 | 18.27% | 15.99% |
| 3,000 | 21.83% | 19.51% |
| 4,000 | 23.72% | 21.65% |
| 5,000 | 25.08% | 23.38% |

As shown in Table 4, even with a much larger batch size, SLIDE and PGHash are unable to crack 30% on Amazon-670K. We would like to note that using a smaller batch size (like the $M = 256$ value we use in our Amazon-670K experiments) resulted in an even further drop in accuracy. These empirical results back-up the notion that SimHash is ill-fit for Amazon-670K.

**Wiki-325K Performance.** In Figure 7, we showcase how PGHash-D performs on Wiki-325K. Quite similar to the Amazon-670K results (shown in Figure 5), PGHash-D almost exactly matches up with SLIDE. In order to map how well our training progresses, we periodically check test accuracies. However, since the test set is very large, determining test accuracies over the entire test set is infeasible due to time constraints on the cluster. Therefore, we determine test accuracies over 30 batches of test data as a substitute as is done in [8, 7]. For Delicious-200K and Amazon-670K the entire test set accuracies matched the randomly sampled batches, however the randomly sampled batches underestimate the true test accuracies for Wiki-325K. For Wiki-325K, the true test accuracy ran about 5% greater than the sampled test accuracy values.

**Matching Full-Training Performance.** Along with the failure for SimHash to perform well on Amazon-670K and Wiki-325K, SLIDE and PGHash(-D) are unable to match the performance of full-training on these data-sets. This is observed empirically for Amazon-670K by GResearch in the following article `https://www.gresearch.co.uk/blog/article/implementing-slide/`. We surmise that the failure of SLIDE and PGHash(-D) to match full-training performance on Amazon-670K and Wiki-325K arises due to the small average labels per point in these two data-sets (5.45 and 3.19 respectively). Early on in training, SLIDE and PGHash(-D) do not utilize enough activated neurons. This is detrimental to performance when there are only a few labels per sample, as the neurons corresponding to the true label are rarely selected at the beginning of training (and these final layer weights are tuned much slower). In full-training, the true neurons are always selected and therefore the final layer weights are better adjusted from the beginning. We also note that [33] requires a hidden layer size of 1024 for a distributed version of SLIDE to achieve improved test accuracies for Amazon-670K. Thus, increasing the hidden layer size may have improved our performance (we kept it as 128 to match the original SLIDE paper [8]).

# B    PGHash: angle versus Hamming distance

In this section, we visually explore the degree to which PGHash is a consistent estimator of angular similarity. Specifically, let $x, y \in \mathbb{R}^d$: then we know by Theorem 1 that $\mathcal{H}^{PG}(c, d)$ is an LSH for $\cos(x_c, y_c)$. We demonstrate that in the unit vector regime, $\theta_c = \arccos(\cos(x_c, Y_c))$ is an acceptable surrogate for $\theta = \arccos(x, Y)$, where $Y = \{y^i\}_{i=1}^N$ and $Y_c = \{y_c^i\}_{i=1}^N$.

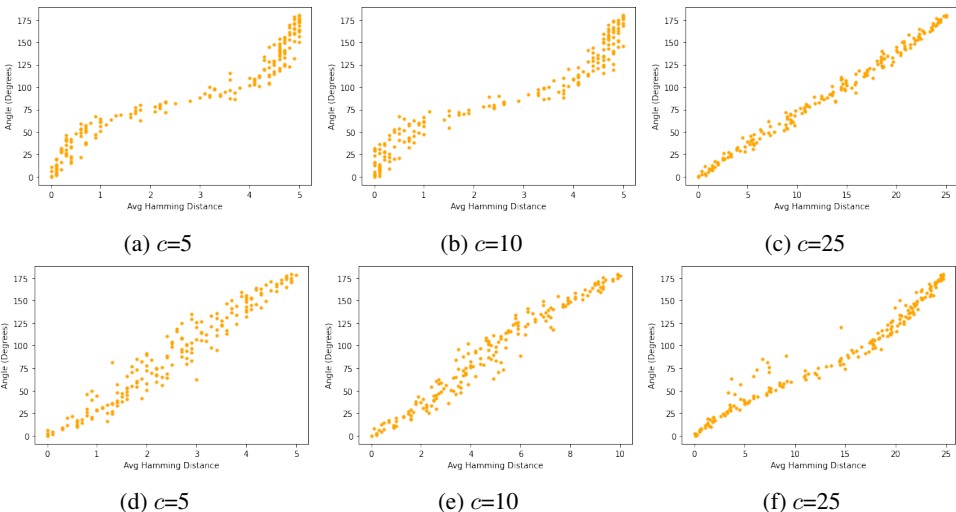

|           |           |           |
|-----------|-----------|-----------|
| (a) $c$=5 | (b) $c$=10 | (c) $c$=25 |
| (d) $c$=5 | (e) $c$=10 | (f) $c$=25 |

Figure 8: **Angle/Hamming Distance as a function of sketch dimension.** The average Hamming distance between a PGHashed fixed unit vector $x \in \mathbb{R}^{100}$ and a collection of vectors $y_i \in \mathbb{R}^{100}$ which form different angles with $x$. Increasing sketch dimension $c$ smooths and reduces the variance of the scatter towards linear correlation. Furthermore, the Hamming scales linearly with $c$, improving discernibility. (a)-(c) & (d)-(f) are independent series.

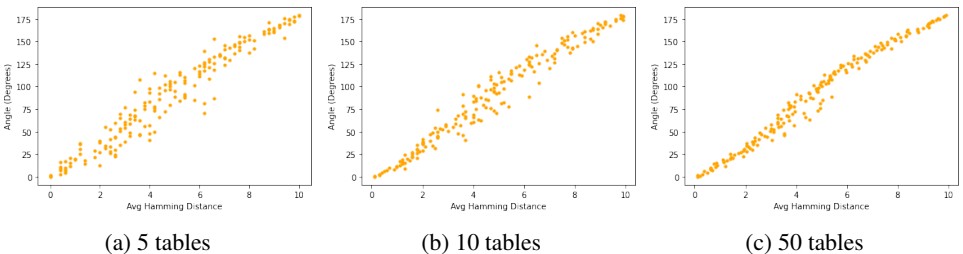

(a) 5 tables         (b) 10 tables         (c) 50 tables

Figure 9: **Angle/Hamming Distance as a function of tables** The average Hamming distance between a PGHashed fixed unit vector $x \in \mathbb{R}^1 00$ and a collection of vectors $y_i \in \mathbb{R}^{100}$ which form different angles with $x$ and fixed sketch dimension $c = 10$. Increasing the number of tables reduces variance.

## C    Additional proofs

**Fact 3.** *Let $x, y, e_1, e_2$ be $d$-dimensional unit vectors such that that the $e_i$ lie on the unit circle contained with the plane spanned by $x$ and $y$ (denoted as $S_{x,y}$) and $e_1 \perp e_2$. Consider the point $v$ on $S_{x,y}$ such that the line through it bisects the angle of the lines passing through $x$ and $y$. Let $\eta = \arccos(\cos(v, e_1))$. Denote $\theta = \frac{1}{2}\arccos(\cos(x, y))$. Then we may write $x = \cos(\eta + \theta)e_1 + \sin(\eta + \theta)e_2$ and $y = \cos(\eta - \theta)e_1 + \sin(\eta - \theta)e_2$.*

### C.1    Proof of Theorem 2

*Proof.* Let $\theta = \frac{1}{2}\arccos(\cos(x, y))$ where $x, y \in \mathbb{S}^{d-1}$ and $A = B^\top B$. The cosine similarity between $x_c = Bx$ and $y_c = By$ (for $B$ correspondent to a $(d, c)$-folding), is expressible as

$$\cos(x_c, y_c) = \frac{x^\top B^\top By}{(x^\top B^\top Bx)(y^\top B^\top By)} = \frac{x^\top Ay}{\sqrt{(x^\top Ax)(y^\top Ay)}}. \tag{1}$$

Consider the SVD $B = UDV^\top$ where $U$ and $V$ are orthogonal and $D$ is $c \times d$ rectangular diagonal matrix. We have then that $A = B^\top B = V\hat{D}^2V^\top$. (Here $\hat{D}$ is now a square diagonal matrix containing squared $D_{ii}$ along the diagonal and 0 everywhere else.) Notice that choice of $U$ nor the ordering of columns $v_i$ of $V$ affects the angle calculation in Equation 1. First, we re-order the columns of $V$ so as to order the diagonal entries $d_i$ of $D$ (i.e., the squared singular values) in decreasing order, and as an abuse of notation set $B = \frac{1}{d_1}DV^\top$. Denoting $\hat{\lambda}_i = d_i/d_1$ for $1 \le i \le n$, we have that $Bv_i = \hat{\lambda}_i e_i$. (By construction of $B$ we have that $d_i \in \{\frac{d}{c}, 0\}$, therefore, $\hat{\lambda}_i \in \{1, 0\}$)

Consider $B$ acting on $S^{d-1}$: it scales each dimension by $\hat{\lambda}_i$, thus (as with any linear transformation of a sphere), transforms it into an ellipsoid, with $c$ principal axes determined by the $v_i$. The greatest possible distance from the origin to the ellipsoid $BS^{d-1}$ is 1 while the shortest possible distance is 0. Now consider the unit circle $S_{x,y} = \{v \in \text{span}(x, y) \; : \; ||v|| = 1\}$. We have that $BS_{x,y} \subset BS^{d-1} \cap BU$ is an ellipse (since the intersection of an ellipsoid and plane is always an ellipse).

Choose unit $w_1$ and $w_2$ belonging to $S_{x,y}$ such that $w_1 \perp w_2$. by By Fact 3, we may parameterize our vectors as $x = \cos(\eta - \theta)w_1 + \sin(\eta - \theta)w_2$ and $y = \sin(\eta + \theta)w_1 + \sin(\eta + \theta)w_2$, where $\eta$ is the angle made with $w_1$ with the bisector of $x$ and $y$. By assumption, $||Bw|| \ge \alpha$ (the minimal shrinking factor of $B$ on $S_{x,y}$) for some positive $\alpha$. Denoting $\lambda = \frac{d}{c}$ (the maximal stretching factor of $B$ on $S_{x,y}$), we have that the angle between $Bx$ and $By$ is upper-bounded by

$$f(\eta) = \arctan(\frac{\alpha}{\lambda}\tan(\eta + \theta)) - \arctan(\frac{\alpha}{\lambda}\tan(\eta - \theta)) \tag{2}$$

.

The numerator of $\frac{df}{d\eta}$ is $\beta(1 - \beta)(1 + \beta)\sin(2\theta)\sin(2\eta)$ where $\beta = \alpha/\lambda$. The derivative is trivially 0 if (1) $\beta = 0$, (2) $\beta = 1$, or (3) $\theta = 0$. (1) will not occur as we assume that $S_{x,y}$ does not contain a 0-eigenvector of $A = B^\top B$. (2) can only occur if $A$ is a multiple of the identity matrix (which it is not by construction), and (3) implies that $x$ and $y$ are parallel, in which case their angle will not be

distorted. Aside from these pathological cases, the critical points occur at $\eta = 0, \pi/2$. We have then that $\cos(Bx, By)$ lives between $\cos(f(0)) = \frac{1 - \beta^2 \tan^2 \theta}{1 + \beta^2 \tan^2 \theta}$ and $\cos(f(\pi/2)) = -\frac{\tan^2 \theta^2 - \beta^2}{\tan^2 \theta^2 + \beta^2}$.

$\square$

**Remark.** The constant $\beta$ has an enormous influence on the bounds in Theorem 2. The smaller the $\alpha$ (i.e., shrinking of $||w||$), the greater the bounds on distortion. Although we have imposed constraints on $x, y$, if we treat them as any possible pair of random unit vectors, then the $w$ in $S_{x,y}$ effectively becomes a random unit vector as well. We can exactly characterize the distribution of $||BX||$ where $X$ denotes a random variable which selects a $d$-dimensional unit vector uniformly at random.

### C.2  Proof of Proposition 1

*Proof.* We can sample a $d$-dimensional vector uniformly at random from the unit sphere $S^{d-1}$ by drawing a $d$-dimensional Gaussian vector with iid entries and normalizing. Let us represent this as the random variable $X = Z'/||Z'||$ where $Z' \sim \mathcal{N}(0, I_d)$. Consider a $(c, d)$-folding matrix $B$, i.e., a $d/c$ horizontal stack of $c \times c$ identity matrices (let us assume $c|d$). We are interested in determining the distribution of $||BX||^2$. For ease of notation, consider the permutation $Z$ of $Z'$ where $Z_i = Z'_{(\lfloor \frac{d/c}{i} \rfloor - 1)*(d/c)+i \pmod{d/c}}$. Since this permutation is representable as an orthogonal matrix $P$ (and multi-variate Gaussians are invariant in distribution under orthogonal transformations), we may instead consider $X := P(Z'/||Z'||)^2 = Z/||Z||^2$. We may write the norm-squared as

$$||BX||^2 = \frac{(Z_1 + \cdots + Z_{d/c})^2}{||Z||^2} + \frac{(Z_{d/c+1} + \cdots + Z_{2d/c})^2}{||Z||^2} + \cdots + \frac{(Z_{(c-1)(d/c)+1} + \cdots + Z_d)^2}{||Z||^2}.$$
(3)

Consider the first term $\frac{(Z_1 + \cdots + Z_{d/c})^2}{||Z||^2}$. First note that for any unit vector $u$, the distribution of $\frac{(u^\top Z)^2}{||Z||^2}$ does not depend on choice of $u$. Consider the unit vector $u'$ then which contains $\sqrt{d/c}$ in the first $d/c$ entries and $0$ otherwise. Then $\frac{(u'^\top Z)^2}{||Z||^2}$ is equivalent to $d/c$ times our first term. Of course, since $\frac{(e_1^\top Z)^2}{||Z||^2}$ has the same distribution as $\frac{(u'^\top Z)^2}{||Z||^2}$, we have by transitivity that $\frac{Z_1^2}{||Z||^2} \overset{d}{=} (n/q)\frac{(Z_1 + \cdots + Z_{d/c})^2}{||Z||^2}$.

By extending the discussion above to the other terms, and by their independence with respect to rotation of $Z$ (since their numerators contain squared sums of mutually disjoint $Z$ coordinates), we have that

$$||BX||^2 \overset{d}{=} \frac{d}{c} \cdot \frac{Z_1^2 + Z_{d/c}^2 + Z_{2d/c}^2 + \cdots + Z_d^2}{||Z||^2}.$$
(4)

The distribution of $\frac{Z_1^2 + Z_{d/c}^2 + Z_{2d/c}^2 + \cdots + Z_d^2}{||Z||^2}$ is well-known to follow a $\text{Beta}(\frac{c}{2}, \frac{d-c}{2})$ distribution [13]. In totality, $||BX||^2 \overset{d}{=} \frac{d}{c}\text{Beta}(\frac{c}{2}, \frac{d-c}{2})$. However, we will move to the four parameter description of this scaled Beta distribution which is $\text{Beta}(\frac{c}{2}, \frac{d-c}{2}, 0, \frac{d}{c})$. The pdf and expected value follows by the usual statistical descriptions of this distribution, which can also be found in [13]. $\square$

Figure 10 depicts how $(d, c)$-foldings affect the norms of unit vectors.

## D  Additional theory

In this section, we provide additional theory relevant to SimHash.

We present several well-known results regarding SimHash.

**Proposition 2** (SimHash estimation). *Let $x, y \in \mathbb{S}$, i.e., unit $d$-dimensional vectors. Denote $\theta = \arccos(|\cos(x, y)|)$. Let $v \in S^d$ be a unit vector drawn uniformly at random (according to the Haar measure, for example). Then,*

$$Pr[sgn(v^\top x) \neq sgn(v^\top y)] = \frac{\theta}{\pi}.$$
(5)

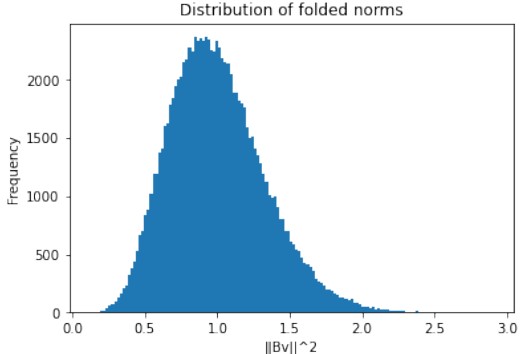

Distribution of folded norms

Figure 10: **Distribution of folded norms.** 100k randomly drawn unit vectors ($d = 128$) are folded down to length 16 by are usual $(d, c)$-folding procedure. Depicted is a binned histogram of the norms. As predicted by the statistical description of $||BX||^2$, where $X$ is a randomly drawn unit vector, the mass is centered at 1, i.e., most norms are preserved. Empirically we observe that folded rarely exceed $\sqrt{\frac{128}{16}}$, although the theoretical support is $[0, 8]$: this concurs with the pdf.

*Proof.* We reproduce the argument of [12]. We have by symmetry that $Pr[\text{sgn}(v^\top x) \neq \text{sgn}(v^\top y)] = 2Pr[v^\top x > 0, v^\top y < 0]$. The set $\mathcal{U} = \{v \in S^d : v^\top x > 0, v^\top y \leq 0\}$ corresponds to the intersection of two half-spaces whose dihedral angle (i.e., angle between the normals of both spaces) is exactly $\theta$. Intersecting with the $d$-dimensional unit sphere produces gives a subspace of measure $\frac{\theta}{2\pi}$, therefore, $2Pr[v^\top x > 0, v^\top y < 0] = \frac{\theta}{\pi}$, completing the argument. $\square$

**Corollary 2.** *Let $v$ instead be a $d$-dimensional random Gaussian vector with iid entries $\sim \mathcal{N}(0, 1)$. Then for $x, y \in \mathbb{R}^d$,*

$$Pr[\text{sgn}(v^\top x) \neq \text{sgn}(v^\top y)] = \frac{\theta}{\pi} \tag{6}$$

*Proof.* Randomly drawn, normalized Gaussian vectors are well-known to be uniformly distributed on the unit sphere. $\square$

In the setup as above, let the $X$ be a random variable which returns 1 if $x$ and $y$ have differing signs when taking the standard inner product with a randomly drawn Gaussian $v$. Let $X_1, X_2, \ldots, X_n$ represent a sequence of independent $X$ events. Then,

**Proposition 3.** $\mathbb{E}[\frac{1}{n} \sum_{i=1}^n X_i] = 1 - \frac{\theta}{\pi}$ *and* $\mathbb{V}[X] = \frac{1}{N} \frac{\theta}{\pi}(1 - \frac{\theta}{\pi})$.

Given that PGHash is equivalent to a SimHash over $(d, c)$-foldings of $R^d$, the variance reduction we observe by using multiple tables (Figure 9 is explainable by Proposition 3.

