# OpenReview forum: "Large-Scale Distributed Learning via Private On-Device LSH"
_NeurIPS.cc/2023/Conference — NeurIPS 2023 poster_

### Official Review · Reviewer_7yAa · 2023-07-05

**Soundness:** 4 excellent
**Presentation:** 4 excellent
**Contribution:** 4 excellent
**Rating:** 7
**Confidence:** 5

**Summary:**

This paper presents a novel framework for on-device locality-sensitive hashing (LSH) that addresses the limitations of existing algorithms in terms of computational efficiency, memory constraints, and privacy concerns. The authors introduce a new family of hash functions that enables each device to generate hash tables independently, without relying on a central host. This approach allows for personalized and private LSH analysis while conserving memory and computational resources.

**Strengths:**

1. The proposed PGhash family is novel. It addresses computational and memory challenges introduced by repeated randomized projection of full-layer weights in distributed settings.

2. The authors claim to have proven several statistical and sensitivity properties of their PGhash functions, ensuring the reliability and robustness of the proposed framework. They also present experimental results that demonstrate the competitiveness of their approach in training large-scale recommender networks compared to existing LSH frameworks that assume unrestricted on-device capacity.

3. The paper is well-written. The theoretical formulation is well-organized.

**Weaknesses:**

NA

**Questions:**

Is there any intuition on the choosing between PGhash version of DWTA and SimHash in practice?

**Limitations:**

The authors adequately addressed the limitations.

---

> ### Author Rebuttal · Authors · 2023-08-10
>
> We thank the reviewer for their valuable feedback. We address all questions and concerns below.
>
> ---
>
> > Is there any intuition on the choosing between PGHash version of DWTA and SimHash in practice?
>
> DWTA and SimHash approximate similarity in fundamentally different ways:
>
> * DWTA computes similarity between vectors based on their relative attributes (similar ordering of indices by magnitude).
> * SimHash approximates cosine similarity between vectors.
>
> In practice, selection of a PGHash version of DWTA versus SimHash stems from the desired speed of LSH analysis as well as the label sparsity of the data.
> 1. Speed: DWTA is faster than SimHash (one of the fastest known hashing schemes) since matrix multiplication is not needed. DWTA can also generate multiple hashes at once (SimHash requires multiple random projections to get multiple hashes).
> 2. Label sparsity: We can further speculate on choice of hash based on some interesting empirical data: Table 4 in our supplemental material demonstrates that SimHash is inferior for training on Amazon-670K, and we also use require usage of DWTA to train over Wiki-325K (Figure 7). The average number of labels per sample are as follows, Delicious-200K: ~75 labels, Amazon-670K: ~5 labels, Wiki-325K: ~3 labels, which suggests that DWTA should be used when the expected number of labels per point is **extremely low.**

---

### Official Review · Reviewer_QaZe · 2023-07-18

**Soundness:** 3 good
**Presentation:** 4 excellent
**Contribution:** 3 good
**Rating:** 5
**Confidence:** 3

**Summary:**

The paper proposes a family of hashing functions that reduces the memory cost. Let W be a weight matrix. It is observed that in many applications, one needs to draw a Gaussian matrix T and compute T * W when constructing hash tables. Also, given a query x, one needs to compute T * x which requires the storage of the entire matrix T. These two steps will be less practical on memory-limited devices. The main contribution of the work is a more efficient hashing scheme. The key idea is to perform a low-dimensional projection B of x on to dimension c, and use a smaller Gaussian matrix S. It is shown that the cosine distance on Bx preserves LSH property.

**Strengths:**

+ The problem and results are clearly presented.

+ The saving of memory is important to many applications.

+ The techniques are easy to follow.

+ A set of experiments were included to support the theory.


**Weaknesses:**

- My primary concern is the novelty of the approach. It turns out that Theorem 1 is the main result of the paper, which shows that projection with the matrix B = [I_c | I_c | ... | I_c] would preserve cosine distance in the sense of LSH. I am however uncertain whether the design of B is novel, or has been considered before in a similar context. (I guess it appeared somewhere but I cannot recount.) I defer to authors' clarification and other reviewers' comments.

- The parameter 'c' controls the new dimension, which is supposed to be lower bounded by some quantity relevant to d or the training size, or will change the performance of the similarity function Sim_c, but there was no such discussion or condition.

- Algorithm 1 looks superficial and lacks guarantee.

**Questions:**

see weakness

---

> ### Author Rebuttal · Authors · 2023-08-10
>
> We thank the reviewer for their valuable feedback. We address all questions and concerns below.
>
> >My primary concern is the novelty of the approach.
>
> To the best of our knowledge, **our LSH approach is novel** due to the following reasons:
> - Existing literature is not focused on the serial generation of pseudo-random projections, namely because there is no memory concern -- SLIDE [R1], Mongoose [R2], [R3], G-SLIDE [R4], etc., are designed for single-machine setups.
> - Projections over federated settings is a relatively underexplored setup -- Federated SLIDE [R5] performs LSH analysis of hashed client data at the server (and therefore, avoids memory constraints), which we seek to avoid due to privacy concerns.
> - Furthermore, our on-device approach allows for the personalization of LSH analysis by allowing devices to select their own LSH hyperparameters (which other approaches lack).
>
> > I am however uncertain whether the design of $B$ is novel, or has been considered before in a similar context.
>
> To the best of our knowledge, the structure of our $B$ matrix is novel in the context of LSH methods. The design of $B$ comes from the consideration of a nontrivial problem:
> - Given a **fixed** $c \times d$ matrix $B$ with $c<<d$, produce a sequence of  $c \times c$ matrices $S_i$ for $1\leq i\leq n$ such that the $S_iB$ *resemble* random $c \times d$ Gaussian matrices. Of course, it is not mathematically possible to obtain a sequence of independent, random Gaussian matrices given a fixed $B$, so we will try to produce near-Gaussian matrices.
>
> It is clear that the $S_i$ should be $c\times c$ random Gaussian matrices. Since a random variable $\mathcal{Z}\sim\mathcal{N}(0,I_c)$ is invariant under orthogonal transformations, we should have $B=[O_1 | O_2 |\cdots | O_{d/c}]$, where the $O_i$ are orthogonal matrices.
>
> We set $O_i=I_c$ since:
> 1. It is an **effective discriminator** of angular similarity according to Figures 2 and 8,
> 2. Its simple structure makes it possible to analyze the distribution of $||Bv||^2$,
> 3. For instances where dissimilar vectors are grouped as similar, it simply results in an increased number of weights to train, which according to Figure 6(a) is **never a significant amount**.
>
> ---
>
> > The parameter 'c' controls the new dimension, which is supposed to be lower bounded by some quantity relevant to d or the training size, or will change the performance of the similarity function Sim_c, but there was no such discussion or condition.
>
> In Proposition 1, the distribution of $||Bv||^2$ critically depends on the sketch dimension $c$. Namely, a higher value of $c$ (i.e., less aggressive compression) produces a generalized Beta distribution with less variance. The altered magnitude of vectors under the multiplication of $B$, i.e., $||Bv||^2$ will affect the extent of angle distortion presented in Theorem 2 (explained in lines 238-243), so the degree of distortion is affected by the sketch dimension $c$.
>
> ---
>
> > Algorithm 1 looks superficial and lacks guarantee.
>
> To the best of our knowledge, no LSH-based dynamic pruning approach, such as SLIDE [R1] or its predecessor MIPS-dropout [R2], G-SLIDE [R3], or Federated SLIDE [R4] contains a guarantee because it is difficult to theoretically establish a convergence rate when the parameters are dropped out in a structured, continually-changing manner as opposed to, say, random dropout.
>
> Similar to these other works, we are focused on the similarity estimation/sensitivity of our LSH strategy. Although this is beyond the purview of our work, a starting point for establishing the convergence of adaptive dropout algorithms is to use the theory established by [R6], which demonstrates that federated training on heterogeneous subnetworks can succeed if the overlap of parameters from round-to-round and client-to-client is significant enough.
>
> ---
>
> Thank you for your review. If we have addressed your questions, we would appreciate it if you would consider updating your score. If any other questions or concerns remain, please let us know.
>
> **References:**
>
> [R1] Chen, Beidi, et al. "Slide: In defense of smart algorithms over hardware acceleration for large-scale deep learning systems." (2020)
>
> [R2] Chen, Beidi, et al. "Mongoose: A learnable lsh framework for efficient neural network training." 2020.
>
> [R3] Spring, Ryan, and Anshumali Shrivastava. "Scalable and sustainable deep learning via randomized hashing." (2017)
>
> [R4] Pan, Zaifeng, et al. "G-SLIDE: A GPU-Based Sub-Linear Deep Learning Engine via LSH Sparsification." (2021)
>
> [R5] Yan, Minghao, et al. "Distributed slide: Enabling training large neural networks on low bandwidth and simple cpu-clusters via model parallelism and sparsity." (2022).
>
> [R6] Zhou, Hanhan, et al. "Federated Learning with Online Adaptive Heterogeneous Local Models." (2022)

---

### Official Review · Reviewer_Q7Pn · 2023-08-01

**Soundness:** 2 fair
**Presentation:** 2 fair
**Contribution:** 1 poor
**Rating:** 3
**Confidence:** 4

**Summary:**

**Main Idea** : If you are using LSH based sparsity, then why not project the weight matrices down using random / even structured projections and then use the LSH.

**Potential applications**: device-based training / inference (resource-constrained) / federated learning.

The authors provide theoretical analyse of proposed PGHash and provide some experiments in single and multi-device settings.

**[UPDATE AFTER REBUTTAL]**
Sorry for the late response. Due to unavoidable reasons, i could not get to this earlier.

1. As agreed by authors on most questions on theoretical results, there were issues in the provided theory. While authors provide quick fixes to the theorems. In order to evaluate these fixes, one would have to reconsider the entire theory again.

2. I do not agree with authors argument in favor of structured matrix B. I still believe it has issues and detailed proof must be provided to support it. I am afraid structured matrix B will kill any best bounds you can give. For example, in the example I gave in the review, how can
one ever cope up with the error caused there. For such as example, the bounds have to be trivial [-1,1] for cosine similarity.

3. I am skeptical with the statement that "The block identity structure solves the following problem: given a fixed $c \times d$ matrix $B$ with $c<<d$, produce a sequence of $c \times c$ matrices $S_i$ for $1\leq i \leq n$ such that the $S_iB$ resemble random $c \times d$ Gaussian matrices".  "Resemble" is not what we are looking for in the theory. **Would request ACs to take a serious look at the theory to confirm its correctness.**

About experiments, I would request the authors to show results after the peak as well to get the full picture.


I will be maintaining my score for the submission in its current form.

**Strengths:**

The idea of using projections before LSH might be useful in practice depending on the sensitivity of the model to the compression.

**Weaknesses:**

See question section.

**Questions:**

**Theoretical soundness of approach and provided theorems.**

1. Theorem 1 does not provide any information on the approximation.

     The statement of theorem 1 is obvious as the similarity is defined as cosine similarity over folded vectors. After folding, PGhash is just Simhash so it is not surprising that it is a LSH on folded vectors

2. Theorem 2 does provide some information. But has issues.

   a). The $\alpha$ is infimum of a set and can be equal to $0$ even if the set itself does not contain $0$. (on a related note, i believe the definition of $\alpha$ can be improved in terms of how the set is written). Is it that $\alpha = inf \\{ ||Bv|| , v \in S_{x,y} \\}$ ? I assume this is what the authors meant.

   b) If $\alpha$ is 0, then the range of distortion is trivially -1,1

   c) Proposition 1 is used to comment on the value of $\alpha$ but it's not exactly clear how they are related, the set used in proposition 1 is that of unit sphere in R^d and that in theorem 2 is linear span of $x$ and $y$ where coefficients are from a unit circle . Clearly the two sets are very different. Also, E||Bu|| = 1 cannot be directly extended to understand the range of $\alpha$ as $\alpha$ is infimum of the $ \\{ ||Bv|| , v \in S_{x,y}\\}$

   d) It is possible that writing issues in this section might have mislead me a bit. It would be good to get clarification from authors.

   e) The structured form of matrix B is actually troublesome for theoretical guarantees. Consider a simple example , $d=20$, $c=5$, the vectors be,  $S = \\{ [x_1, 0,0,0,0, x_2, 0,0,0,0, x_3, 0,0,0,0, x_4, 0,0,0,0] | x_1+x_2+x_3+x_4 = 1 \\}$.  Note that two vectors from S can have cosine similarities that vary from $-1$ to $1$ but under PGHash , they will always have similarities of $1$.



**Experiments:**

1. Both from Figure 5 and Figure 4, it seems like higher number of devices causes PGHash to perform worse. Is there any reason why this occurs?
2. I do expect that using projections to reduce size of weight matrices is going to lose accuracy. However, the hope for this paper is that the trade-off only significantly kicks in after large amounts of compression.

    a. The plots provided for Amazon-670K and Delicious-200K are not shown to convergence. So it is impossible to know what the eventual loss of accuracy is.

    b. With multi-device settings, the plots already show that PGhash is losing accuracy as compared to baseline. Given that multi-device setting is the proposed application area of PGhash, it is unclear if the overall proposal has value.

**Limitations:**

Yes.

---

> ### Author Rebuttal · Authors · 2023-08-10
>
> We thank the reviewer for their valuable feedback. Below, we address all questions and concerns.
>
>
> ## Theory
>
>
> > [Q1]: Misleading impression of Theorem 1 being the main result
>
> Theorem 1 and its corollary are warmup results intended to illustrate a simple connection between traditional SimHash and our PGHash, while Theorem 2 and Proposition 1 are our core and novel contributions. In our revision, we rename Theorem 1 as Fact or Proposition.
>
> ---
>
> > [Q2(a)]: $\alpha$ in Theorem 2 might be 0
>
> Indeed, our intended definition is $\alpha=\inf\bigl(||Bv||^2: v\in S_{x,y} \bigr)$. The observation that $\alpha$ might be 0 within our current definition is true and a fixable caveat in our formulation. To remedy this, we now assume that $\alpha=\inf\bigl(||Bv||: v\in S_{x,y} \bigr)>0$, which trivially absorbs assumption of no 0-eigenvectors in $S_{x,y}$ and eliminates the possibility of infinitesimally-vanishing norm. This updated definition and assumption of $\alpha$ will be included in the revision.
>
> > [Q2(b)]: If $\alpha$ is 0, then the range of distortion is [-1,1]
>
> Correct, if $\alpha$ is 0 then the range of distortion is trivially [-1,1]. We assume that $\alpha>0$ to provide meaningful distortion bounds in Theorem 2. Empirically, grouping dissimilar vectors as similar is statistically rare for randomly drawn unit vectors (Figures 2 and 8). **Furthermore, in practice it doesn't hinder training**: this simply results in less pruning since more weight vectors are deemed similar to the input and therefore updated.
>
> ---
>
> > [Q2(*c*)]: Relation of Proposition 1 to $\alpha$
>
> First, we clarify that $S_{x,y}$ is defined as the set $\bigl( v\in \mathrm{span}(x,y): ||v||=1\bigr)$. This was a technical typo that should resolve much of the confusion on relating Proposition 1 and Theorem 2.
>
> It is useful to know how the norms of arbitrary unit vectors are distorted by $B$, which, according to Proposition 1, follows a generalized beta distribution with concentrated mass around 1 (see Figure 10 for a simulation). Indeed, $S_{x,y}$ is not the unit sphere, but since $x$ and $y$ are assumed to be random unit vectors ($\alpha>0$ is assumed), Proposition 1 provides intuition on how we probabilistically expect random, unit linear combinations of $x$ and $y$ to shrink/stretch under $B$.
>
> ---
> > [Q2(e)]: Justifying the efficacy and structure of $B$
>
> The $B$ matrix we suggest,
> 1. Is an **effective discriminator** of angular similarity according to Figures 2 and 8.
> 2. Its **simple structure** makes it possible to analyze the distribution of $||Bv||^2$.
> 3. For instances where dissimilar vectors are grouped as similar (like the example you provided), it simply results in an increased number of weights to train, which according to Figure 6(a) is **never a significant amount.**
>
> The block identity structure solves the following problem: given a **fixed** $c \times d$ matrix $B$ with $c<<d$, produce a sequence of  $c \times c$ matrices $S_i$ for $1\leq i \leq n$ such that the $S_iB$ *resemble* random $c \times d$ Gaussian matrices.
>
> It is clear that we should have $S_i\sim \mathcal{N}(0,I_c)$. Since the standard normal distribution is invariant under orthogonal transformations, $B$ should contain blocks of orthogonal matrices, i.e., $B=[O_1|O_2|\cdots|O_{d/c}]$, where the $O_i$ are orthogonal matrices. Since the $B$ and $O_i$ are fixed, for any Gaussian $S$ the blocks $\{SO_i\}$ are always correlated, so examples of dissimilar vectors grouped as similar **will always exist** and thus we simply choose $O_i=I_c$.
>
> ---
>
> ## Experiments
>
> > [Q2(b)] Novelty of PGHash and performance versus baselines
>
> The novelty of our work is that PGHash is the first:
> 1. On-device and memory-efficient LSH approach,
> 2. Privacy-preserving and personalizable LSH method in the distributed setting.
>
> As mentioned by the reviewer, it is common to "expect that using projections to reduce size of weight matrices is going to lose accuracy". Even still, we empirically showcase:
> * PGHash matches (Figure 5 for Amazon) or is competitive with full-memory baselines **using only 6.25% of the full weight matrix**.
> * PGHash matches baselines while allowing on-device, private LSH analysis with **personal compression rates and hash code lengths** (something that has never been allowed previously).
>
> ---
>
> > [Q1] Performance with higher numbers of devices worse than single device?
>
> Slightly degraded performance with more devices is typical and expected in FL since we are averaging gradients trained on partitions of the dataset. This is sub-optimal compared to optimizing over the entire dataset (especially with large datasets like ours). Even still, final accuracy for Delicious degrades by a negligible amount (<1%) in the FL setting. Nonetheless,
> - Peak accuracy should also be taken into consideration when evaluating performance, and multi-device achieves a **higher** peak accuracy on Delicious.
> - In contrast to Delicious, multi-device **performs better** on Amazon for both peak and final accuracy (+4.5% as shown in Figure 5).
>
> > [Q2(a)] Convergence of Amazon and Delicious accuracies
>
> Peak accuracies for our given architecture (with 1 device) are approximately 45% and 33% for Delicious and Amazon respectively. These are the values within SLIDE [R1] and validated within our own work. For Delicious200K, peak accuracies are reached between 2-3 thousand iterations. This is why [R1] only plots the first 3,000 iterations within their experiments. For Amazon, we ended our accuracy plots after peak distributed accuracy is achieved. What is important to show, which we do in Figures 4 and 5, is that **PGHash achieves peak distributed accuracy for Delicious and Amazon**.
>
> ---
>
> Thank you for your review. If we have addressed your questions, we would appreciate it if you would consider updating your score. If any other questions or concerns remain, please let us know.
>
> **References:**
>
> [R1] Chen, et al. "Slide: In defense of smart algorithms..." 2020.

---

### Author Rebuttal · Authors · 2023-08-10

### Summary of Rebuttal:

**General Comments:** We thank the reviewers for their valuable feedback and questions on theory and experiments.

**Reviewer highlights:**

Reviewer 7yAa believes that our "approach allows for **personalized** and **private** LSH analysis while conserving **memory** and **computational** resources," "the proposed PGhash family is **novel**, and that "the theoretical formulation is **well-organized**."

Reviewer QaZe remarks "the saving of memory is important to **many applications**" and that our "techniques are **easy** to follow."

Reviewer QZPn notes that our approach has several **potential applications** while also mentioning that "the idea of using projections before LSH might be **useful in practice** depending on the sensitivity of the model to the compression."

**Summary of Core Contributions:** We summarize our key contributions as follows:
* We introduce a **novel** LSH family and framework for **dynamic pruning** of a massive final layer weight in a distributed setting. Our approach allows for serial generation of hash tables, enabling **memory** and **computationally efficient** on-device LSH analysis. Appealing to federated learning principles, PGHash-based LSH analysis is **personalizable** and **private**.
* Our novel LSH family, PGHash, estimates angular similarity between "foldings"/deterministic hashings of the weight vectors and the layer input. We theoretically establish the sensitivity and statistical properties of PGHash, and empirically demonstrate it is a **good discriminator** of similarity over randomly-drawn unit vectors.
* Our framework enables **competitive multi-device training** of a network on extreme multi-label datasets, Delicious-200K and Amazon-670K. Multi-device PGHash training:
    1.  **Matches** federated SLIDE training [1,5,6] of Amazon-670K,
    2.  **Outperforms** single-device training of Amazon-670K,
    3.  **Closely matches** the peak accuracies of SLIDE and federated SLIDE training on Delicious-200K.

    The results above are accomplished all while storing **only 6.25%** of the final massive layer weight (a reduction of **tens of millions of parameters compared to SLIDE**).

**Summary of Changes**: Guided by the helpful suggestions of our reviewers, our changes are as follows:
* Amended definition and assumption of $\alpha$ to improve readability and eliminate an edge case ($\alpha=0$) pertinent to Theorem 2. Core result and proof of Theorem 2 **are preserved.**
* Corrected technical typo in the formulation of $S_{x,y}$ to stress that it should contain unit vectors. This correction is needed to appropriately link Proposition 1 and Theorem 2. Core result and proof of Theorem 2 + Proposition 1 **are preserved**.

**References:**

[R1] Chen, Beidi, et al. "Slide: In defense of smart algorithms over hardware acceleration for large-scale deep learning systems." 2020.

[R2] Chen, Beidi, et al. "Mongoose: A learnable lsh framework for efficient neural network training." 2020.

[R3] Spring, Ryan, and Anshumali Shrivastava. "Scalable and sustainable deep learning via randomized hashing." 2017.

[R4] Pan, Zaifeng, et al. "G-SLIDE: A GPU-Based Sub-Linear Deep Learning Engine via LSH Sparsification." 2021.

[R5] Yan, Minghao, et al. "Distributed slide: Enabling training large neural networks on low bandwidth and simple cpu-clusters via model parallelism and sparsity." 2022.

[R6] Xu, Zhaozhuo, et al. "Adaptive Sparse Federated Learning in Large Output Spaces via Hashing." 2022.

---

### Author Response · Authors · 2023-08-17

Dear Reviewers,

We greatly appreciate your feedback. We have addressed all questions and concerns in our rebuttal. We would appreciate it if the reviewers can update their scores accordingly. Please let us know if you have more comments or questions.

---

### Decision · Program_Chairs · 2023-09-21

**Decision:**

Accept (poster)

**Comment:**

The paper proposes a novel on-device LSH based approach for large scale distributed learning.  Reviewers generally liked the approach and fact that the paper solved a very important and real problem with a technique that seems truly scalable and practical. One reviewer did pointed some concerns with the formalism as well as some other baselines alternatives. The rebuttal  did address the concerns on writing.  It is clear that the proposed methodology is competitive with most existing works and solves the distributed learning while retaining the important piece of privacy, using very clever stitching of existing ideas. The formalism presented are also interesting.  As a result, the paper does cross the bar for publication.